# Bond-Orbital-Resolved Piezoelectricity in Sp^2^-Hybridized Monolayer Semiconductors

**DOI:** 10.3390/ma15217788

**Published:** 2022-11-04

**Authors:** Zongtan Wang, Yulan Liu, Biao Wang

**Affiliations:** 1School of Aeronautics and Astronautics, Sun Yat-sen University, Shenzhen 518000, China; 2Sino-Franch Institute of Nuclear Engineering and Technology, Sun Yat-sen University, Zhuhai 519082, China

**Keywords:** sp^2^-hybridized monolayer, bond-orbital-resolved piezoelectricity, geometric phase, quantum phase transition, valley Hall effect, π-electron piezoelectric engineering

## Abstract

Sp^2^-hybridized monolayer semiconductors (e.g., planar group III-V and IV-IV binary compounds) with inversion symmetry breaking (ISB) display piezoelectricity governed by their σ- and π-bond electrons. Here, we studied their bond-orbital-resolved electronic piezoelectricity (i.e., the σ- and π-piezoelectricity). We formulated a tight-binding piezoelectric model to reveal the different variations of σ- and π-piezoelectricity with the ISB strength (Δ). As Δ varied from positive to negative, the former decreased continuously, but the latter increased piecewise and jumped at Δ=0 due to the criticality of the π-electrons’ ground-state geometry near this quantum phase-transition point. This led to a piezoelectricity predominated by the π-electrons for a small |Δ|. By constructing an analytical model, we clarified the microscopic mechanisms underlying the anomalous π-piezoelectricity and its subtle relations with the valley Hall effect. The validation of our models was justified by applying them to the typical sp^2^ monolayers including hexagonal silicon carbide, Boron-X (X = N, P, As, Ab), and a BN-doped graphene superlattice.

## 1. Introduction

Piezoelectricity, a linear electromechanical intercoupling in non-centrosymmetric dielectrics, is an intriguing subject in solid state physics and plays an important role in various technological applications. Piezoelectric polarization of a crystal contains ionic and electronic contributions. The former stems from the internal ionic displacement caused by strain, while the latter is described by the strain-induced geometric (or Berry) phase shift of the ground-state wavefunction [1,2,3]. Specifically, the electronic piezoelectric coefficient is determined by the Brillouin zone (BZ) integration of the piezoelectric Berry curvature (PBC) defined in the parameter space containing strain [4,5,6]. The PBC formally resembles the momentum space Berry curvature (MBC) [7], which is crucial for understanding many geometric phase phenomena such as the quantum Hall effect [8] and quantum phase transition (QPT) [9,10,11], etc. Moreover, encoded with the information of the ground state, electronic piezoelectricity may also serve in detecting the QPT characterized by a Berry phase jump [12,13,14]. It is therefore worth investigating the mechanisms of electronic piezoelectricity and its correlation with other geometric phase phenomena.

Technically, nanomaterials with excellent piezoelectricity are pursued for next-generation electronic devices. Recent breakthroughs in two-dimensional (2D) piezoelectricity offer new opportunities in this respect. Many 2D materials [15,16,17,18,19,20,21,22,23,24,25,26] such as 2D transition metal dichalcogenides [16], graphitic carbon nitride [18], and hexagonal boron nitride [19], have been theoretically predicted or experimentally confirmed to be piezoelectric. Among them, sp^2^-hybridized monolayers; e.g., planar group III-V [19,20,21,22,23,24] and IV-IV [24,25,26] binary compounds, have attracted increasing attention due to their bright application prospects in piezotronics and theoretical importance in understanding 2D piezoelectricity.

Sp^2^ piezoelectrics exhibit superior electro-mechanical properties that are deeply rooted in their peculiar sp^2^ hybridization: the strong σ-bonds construct a robust honeycomb lattice that can withstand a large deformation [27]; the π-orbitals form the gapped low-energy dispersion with two non-equivalent valleys, mimicking the behavior of massive Dirac fermions [28,29,30], and an elastic deformation can cause the two valleys to drift along opposite directions, then emerging as pseudo-gauge fields [31]. Due to the inversion symmetry breaking (ISB), they are piezoelectric. The ISB strength and the bond polarity of the sp^2^ piezoelectrics are determined by the electronegativity difference between their two constituent atoms and are tunable by altering the combination of atoms [27]. Physically, their electronic piezoelectricity is contributed by both σ- and π-bond electrons. We thus naturally asked (i) what respective roles the two contributions play in the piezoelectricity and (ii) how they vary with the ISB strength or bond polarity. The ISB also endows the sp^2^ monolayers with a non-zero MBC, resulting in the valley Hall effect [32]. Given the geometric essence of electronic piezoelectricity, it is unclear how the non-zero MBC manifests itself in the piezoelectricity of sp^2^ monolayers. In addition to intrinsic piezoelectricity, an engineered piezoelectricity can also be induced by ISB in centrosymmetric sp^2^ crystals such as graphene [33]. Several schemes of this kind for piezoelectric engineering in sp^2^ materials have been proposed [34,35,36], but further guidelines beyond this ISB criterion are still lacking.

In this work, by combining the tight-binding (TB) approximation and the modern theory of polarization, we systematically explored the bond-orbital-resolved piezoelectricity of the π- and σ-electrons (i.e., the π- and σ-piezoelectricity) in sp^2^ semiconductors. Firstly, by elaborating a TB piezoelectric model for a prototypical sp^2^ crystal, we revealed the different variations in the σ- and π-piezoelectricity with the ISB-strength (Δ). As Δ varied from positive to negative, the former decreased continuously; whereas the latter increased piecewise and showed a discontinuity at Δ=0 due to the criticality of the π-electrons’ ground-state geometry during the QPT. Thus, the electronic piezoelectricity near Δ=0 was predominated by the π-electrons. Then, the microscopic mechanisms of anomalous π-piezoelectricity and its subtle relation with the valley Hall effect were clarified using an analytical model. Finally, we validated our models by applying them to typical sp^2^ materials: hexagonal silicon carbide (h-SiC), Boron-X (h-BX, X = N, P, As, Ab), and a BN-doped graphene (BNG) superlattice. The piezoelectric engineering of the sp^2^ materials and relevant strategies are also discussed.

## 2. Methods and Formulas

### 2.1. Strain-Dependent TB Hamiltonian for Sp^2^ Piezoelectric Crystals

We began with the nearest-neighbor TB Hamiltonian for unstrained sp^2^ piezoelectrics shown in Figure 1a, which, by neglecting the spin–orbit coupling, can be written as [37,38,39]:(1)$$H=\sum_{i_{A},\mu }\epsilon_{\mu }^{A}c_{i_{A},\mu }^{†}c_{i_{A},\mu }+\sum_{i_{B},\mu }\epsilon_{\mu }^{B}c_{i_{B},\mu }^{†}c_{i_{B},\mu }+\sum_{i,\delta_{I},\mu ,\mu^{\prime }}t_{\mu ,\mu^{\prime }}^{I}\left(c_{i,\mu }^{†}c_{i+\delta_{I},\mu^{\prime }}+h.c.\right)$$
where the operator $c_{i_{A},\mu }^{†}(c_{i_{B},\mu })$ creates (annihilates) an electron in the atomic orbital $\mu \in \{s,p_{x},p_{y},p_{z}\}$ located at the A(B) sublattice site $i_{A}(i_{B}=i_{A}+\delta_{I})$ with the on-site energy $\epsilon_{\mu }^{A(B)}$, and $t_{\mu ,\mu^{\prime }}^{I},(I=1,2,3)$ represents the hopping from orbital μ to orbital $\mu^{\prime }$ along the three nearest-neighbor vectors $\delta_{1}=(0,-1)a$, $\delta_{2}=(\sqrt{3},1)a/2$, and $\delta_{3}=(-\sqrt{3},1)a/2$, with a being the unstrained bond length. The on-site energy differences $\Delta_{\mu }=\epsilon_{\mu }^{A}-\epsilon_{\mu }^{B}$ reflect the ISB strength and acted as the control parameters in our following piezoelectric model. Under an in-plane strain ε, the nearest-neighbor vectors transform as $\delta_{I}^{\prime }=\delta_{I}+\varepsilon \cdot \delta_{I}$ if assuming the clamped-ion approximation [5]. Accordingly, within the Slater–Koster framework [37], the strain-dependent hopping $t_{\mu ,\mu^{\prime }}^{I}(\varepsilon )$ can be expressed in terms of the two-center integrals $V_{\chi }(\varepsilon )=V_{\chi }^{0}\exp(1-\beta_{\chi }|\delta_{I}^{\prime }|/a)$ as shown in Table 1, where $V_{\chi }^{0}$ (χ=ssσ,spσ,ppσ, or ppπ) is the two-center integral for unstrained lattice and $\beta_{\chi }$ is the electron-lattice coupling parameter [40]. Then, the strain-dependent TB Hamiltonian can be straightforwardly constructed by replacing the hopping terms in Equation (1) with the strain-modified ones ($t_{\mu ,\mu^{\prime }}^{I}(\varepsilon )$).

For a uniform strain reserving the in-plane reflection symmetry; i.e., $n_{z}=0$, the π-bond orbital $p_{z}$ does not couple with the orbitals forming the σ-bond $\overset{⌢}{\mu }\in \{s,p_{x},p_{y}\}$ because the corresponding hoppings $t_{\overset{⌢}{\mu },p_{z}}^{I}(\varepsilon )$ given in Table 1 vanish. Accordingly, the eight bands of sp^2^ piezoelectrics are decoupled into the π- and σ-bands. This can be seen explicitly in the ***k***-space strain-dependent Hamiltonian, which in the basis $\{s^{A},p_{y}^{A},p_{x}^{A},s^{B},p_{y}^{B},$ $p_{x}^{B}\}⊕\{p_{z}^{A},p_{z}^{B}\}$ takes the following block-diagonal form:(2)$$H(k,\varepsilon )=\left[\begin{array}{cc} H^{\sigma }(k,\varepsilon ) & 0\\ 0 & H^{\pi }(k,\varepsilon ) \end{array}\right]$$
The element of the 6×6 matrix $H^{\sigma }(k,\varepsilon )$ for σ-bands reads:(3)$$H_{mn}^{\sigma }(k,\varepsilon )={\left[H_{nm}^{\sigma }(k,\varepsilon )\right]}^{*}=\sum_{I=1}^{3}t_{\mu_{m},\mu_{n}}^{I}(\varepsilon )e^{-ik\cdot \delta_{I}},(n>m);H_{mm}^{\sigma }=\epsilon_{m},$$
where $\mu_{m(n)}\in \{s^{A},p_{y}^{A},p_{x}^{A},s^{B},p_{y}^{B},p_{x}^{B}\}$ with *m* or *n* ranging from 1 to 6; and $\epsilon_{1}=\epsilon_{s}^{A}$, $\epsilon_{2}=\epsilon_{3}=$ $\epsilon_{p}^{A}$, $\epsilon_{4}=\epsilon_{s}^{B}$, and $\epsilon_{5}=\epsilon_{6}=\epsilon_{p}^{B}$ are the on-site energies. The remaining 2×2 matrix for π-bands is given by:(4)$$H^{\pi }(k,\varepsilon )=\left[\begin{array}{cc} \epsilon_{p}^{A} & f(k,\varepsilon )\\ f^{*}(k,\varepsilon ) & \epsilon_{p}^{B} \end{array}\right]$$
where $f(k,\varepsilon )=\sum_{I=1}^{3}t^{I}(\varepsilon )e^{-ik\cdot \delta_{I}}$ with $t^{I}(\varepsilon )=t_{p_{z},p_{z}}^{I}(\varepsilon )$ and $t^{I}(0)=t$ for simplicity.

### 2.2. General Formulas for Electronic Piezoelectricity

Piezoelectricity can be regarded as “unquantized” charge pumping driven by an adiabatic lattice deformation ε(t) evolving along an open-path t∈[0,T]. The polarization difference accumulated during this evolution is related to the piezoelectric current $\tilde{J}$ via the continuity equation $\Delta P=\int_{0}^{T}dt\tilde{J}(t)$. For the leading order, the piezoelectric response is characterized by the rank-3 piezoelectric tensor properly defined as $e_{ijk}=(\partial {\tilde{J}}_{i}/\partial {\dot{\varepsilon }}_{jk}){|}_{\varepsilon \to 0}$ [3]. For the clamped-ion model in the present work, only the piezoelectric current of electrons was relevant in this definition, so henceforth, $e_{ijk}$ will refer to the electronic or clamped-ion piezoelectric tensor unless otherwise specified. In the framework of semiclassical wave packet dynamics, the piezoelectric current is given by [4,5,6,7]:(5)$${\tilde{J}}_{i}=2e\sum_{jk}\sum_{n}\int_{BZ}\frac{dk}{{\left(2\pi \right)}^{2}}\Omega_{k_{i},\varepsilon_{jk}}^{n}(k,\varepsilon ){\dot{\varepsilon }}_{jk}$$
where e is the charge of the electron, the factor of 2 accounts for the spin degeneracy, and $\Omega_{k_{i},\varepsilon_{jk}}^{n}(k,\varepsilon )=i[\langle \partial_{k_{i}}u_{n}|\partial_{\varepsilon_{jk}}u_{n}\rangle -\langle \partial_{\varepsilon_{jk}}u_{n}|\partial_{k_{i}}u_{n}\rangle ]$ is the PBC built from the strained Bloch eigenstate $|u_{n}(k,\varepsilon )\rangle$ of the *n*th occupied valence bands. Consequently, the piezoelectric tensor, by definition, is obtained as [5,6]:(6)$$e_{ijk}=2e\sum_{n}\int_{BZ}\frac{dk}{{\left(2\pi \right)}^{2}}\Omega_{i,jk}^{n}(k)$$
with $\Omega_{i,jk}^{n}(k)=\underset{\varepsilon \to 0}{\lim}\Omega_{k_{i},\varepsilon_{jk}}^{n}(k,\varepsilon )$.

Equation (6) presents a geometric interpretation of the electronic piezoelectricity; the PBC formally resembles the MBC $\Omega_{n}(k)=i[\langle \partial_{k_{x}}u_{n}|\partial_{k_{y}}u_{n}\rangle -\langle \partial_{k_{y}}u_{n}|\partial_{k_{x}}u_{n}\rangle ]$ in the TKNN formula $\sigma_{H}=(e^{2}/\hbar )\sum_{n}\int_{BZ}dk\Omega_{n}(k)/{\left(2\pi \right)}^{2}$ [8]. However, they differ from each other in two aspects. Firstly, the MBC is defined on the compact BZ torus $\left(k_{x},k_{y}\right)$, while the PBC is defined in a non-compact space $\left(k_{i},\varepsilon_{jk}(t)\right)$ as topologically different from the BZ torus. This results in the difference between their ***k***-integral $\sigma_{H}^{n}$ and $e_{ijk}^{n}$ for the isolated *n*th band: $\sigma_{H}^{n}$ must be quantized [8], but $e_{ijk}^{n}$ is not necessarily quantized. Secondly, in the presence of time-reversal symmetry (TRS) or inversion symmetry (IS), their ***k***-space distributions show contrary parities. In systems with TRS, the MBC is an odd function of ***k***, leading to a vanishing $\sigma_{H}^{n}$ [7]; in contrast, the PBC is an even function of ***k***, thus admitting a non-zero $e_{ijk}^{n}$ [4]. For systems with IS, the MBC is an even function of ***k***, while the PBC is an odd function of ***k***, the latter of which requires that $e_{ijk}^{n}=0$ and, as expected, excludes the piezoelectricity in such systems. Despite these differences, there can be subtle relations between them when the local patches of $\left(k_{i},\varepsilon_{jk}(t)\right)$ and $\left(k_{x},k_{y}\right)$ can be linearly mapped into each other (see Section 3.2).

### 2.3. Details of DFT Computation

To fit the TB parameters for the sp^2^ crystals considered in Section 3.3.1, their band structures were calculated within the density functional theory (DFT) using the VASP package [41]. In the calculation, the exchange–correlation effects were treated at a GGA-PBE level, the electron–ion interactions were described by the projector augmented plane-wave method, and the energy cutoff for basis-set expansion was chosen to be 600 eV. For structure optimization, the total energy was convergent within $10^{-7}$ eV, and a ***k***-point mesh of 24×24×1 was used. To exclude the interactions between the neighboring layer, a vacuum layer with a thickness of 20 Å was applied. The processes of fitting TB parameters for these unstrained sp^2^ crystals are presented in Appendix A.

## 3. Results and Discussions

Since piezoelectricity results from ISB, one might intuitively anticipate that it is positively related to the strength of ISB and thus will decrease to zero when the ISB strength decays [24]. However, we will show in this section that this is not the real case for the piezoelectricity in sp^2^ crystals.

### 3.1. Bond-Orbital-Resolved Piezoelectricity in Generic Sp^2^ Crystals

Armed with the above analysis, we will now use the established formulas to study the electronic piezoelectricity of generic sp^2^ crystals. The decoupling between σ- and π-bands allows splitting of the electronic piezoelectric tensor $e_{ijk}$ into the bond-orbital-resolved contributions $e_{ijk}^{\sigma (\pi )}=2e\int_{BZ}dk\Omega_{i,jk}^{\sigma (\pi )}(k)/{\left(2\pi \right)}^{2}$. If indexing the σ- and π-valence (conduction) bands by n=1,2,3 (m=4,5,6) and n=7 (m=8), respectively, the corresponding PBCs, for convenience of calculation, can be written in the following forms [7]:(7)$$\Omega_{i,jk}^{\sigma }(k)=\sum_{n=1}^{3}\Omega_{i,jk}^{n}(k)=i\sum_{n=1}^{3}\sum_{m=4}^{6}\frac{\langle u_{n}^{0}|v_{i}^{\sigma }|u_{m}^{0}\rangle \langle u_{m}^{0}|{\tilde{v}}_{jk}^{\sigma }|u_{n}^{0}\rangle -c.c.}{{\left(E_{n}^{0}-E_{m}^{0}\right)}^{2}}$$
(8)$$\Omega_{i,jk}^{\pi }(k)=\Omega_{i,jk}^{7}(k)=i\frac{\langle u_{7}^{0}|v_{i}^{\pi }|u_{8}^{0}\rangle \langle u_{8}^{0}|{\tilde{v}}_{jk}^{\pi }|u_{7}^{0}\rangle -c.c.}{{\left(E_{7}^{0}-E_{8}^{0}\right)}^{2}}$$
where $E_{n(m)}^{0}(k)$ and $|u_{n(m)}^{0}(k)\rangle$ are the eigenenergy and eigenstate of the strain-independent Hamiltonian H(k), respectively; $v_{i}^{\sigma (\pi )}(k)=\partial_{k_{i}}H^{\sigma (\pi )}(k)$ is the crystal velocity, ${\tilde{v}}_{jk}^{\sigma (\pi )}(k)=\partial_{\varepsilon_{jk}}H^{\sigma (\pi )}(k,\varepsilon ){|}_{\varepsilon \to 0}$; and c.c. denotes the complex conjugates. The D_3h_ symmetry of sp^2^ crystals requires that $e_{ijk}$ has only one independent component; e.g., $e_{222}$ [15]. Hence, in the following, we will only calculate $e_{222}^{\sigma }$ and $e_{222}^{\pi }$.

Before diving into the calculation, we will briefly explain the parameter setting. Despite the TB parameters for realistic sp^2^ crystals that rely on the material details, their similarity in band structures allows for a unified treatment of their piezoelectricity. Instead of modeling any special sp^2^ crystal, here we are mainly concerned with the general variation trends in $e_{222}^{\sigma (\pi )}$ with the ISB strength parametrized by the on-site energy differences $\Delta_{\mu =s,p}$. Roughly speaking, the $\Delta_{\mu }$ are positively related to the electronegativity difference between A and B atoms [42]; for a qualitative investigation, it was adequate to assume $\Delta_{p}=\Delta_{s}=\Delta$. What we wanted to determine was how $e_{222}^{\sigma (\pi )}$ varied with Δ. To focus on this main motif, we further assumed that the hopping terms did not change with Δ. In our TB calculations, we adopted the typical TB parameters $V_{\chi }^{0}$ and $\epsilon_{\mu }$ fitted for unstrained graphene in [39] and $\beta_{\chi }$ fitted for strained graphene in [40], then modified the onsite energies as $\epsilon_{\mu }^{A(B)}=\epsilon_{\mu }\pm \Delta /2$ to reflect the IBS strength. In other words, we took the “ISB-graphene” as a prototype of piezoelectric sp^2^ crystals in view of their similar band structure. In doing this, we did not expect our qualitative results to depend on such a parameter choice.

The calculated $e_{222}^{\sigma }$ and $e_{222}^{\pi }$ as functions of Δ are plotted in Figure 2a. As the figure shows, the magnitude of $e_{222}^{\pi }$ was overall much larger than that of $e_{222}^{\sigma }$ in a quite large range of Δ. This coincided with the previous DFT prediction for the typical sp^2^ crystal h-BN that the π-electrons would dominate its electronic piezoelectricity [43]. More interestingly, $e_{222}^{\sigma }$ and $e_{222}^{\pi }$ showed opposite variation trends with the ISB strength: as |Δ| decayed, $e_{222}^{\sigma }$ decreased and approached zero as intuitively expected, but $e_{222}^{\pi }$ increased anomalously and finally saturated at a giant finite value. This prominent difference between them implied that sp^2^ crystals with a very small ISB would have a quite strong (rather than weak) piezoelectricity contributed almost entirely by the π-electrons, since $e_{222}^{\sigma }$ is neglectable near Δ=0. Remarkably, in contrast to the continuous variation in $e_{222}^{\sigma }$, $e_{222}^{\pi }$ showed an abrupt jump when crossing the peculiar point at Δ=0. Since piezoelectricity is just the strain-induced geometric phase shift of the ground-sate wavefunction, such discontinuity reflected the non-analyticity in the geometry of the π-electron ground-state [13].

In the context of QPT [9,10,11], the non-analyticity of the ground-state geometric phase accompanying gap-closing is a hallmark of the quantum criticality. The critical points marked by gap-closing divide the Hamiltonian parameter space into several regions. Two ground states lying in the same region can be connected by an adiabatic (well-gapped) parameter path along which the observable ground-state properties vary smoothly [9]. However, when the system’s Hamiltonian varies across the critical point that separates two different regions, it will undergo a QPT that is characterized by the “critical behavior”; i.e., an abrupt change in properties related to the ground-state geometry such as the Hall conductance [8] and the Born effective charge [13]. The distinct variation trends in $e_{222}^{\sigma }$ and $e_{222}^{\pi }$ in Figure 2a can also be understood within the framework of QPT. To see this more clearly, let us focus on the expression of the PBC in Equations (7) and (8). The denominators ${\left[E_{m}^{0}-E_{n}^{0}\right]}^{2}$ indicate that $\Omega_{2,22}^{\sigma (\pi )}$ tends to increase as the occupied $E_{n}^{0}$ and unoccupied $E_{m}^{0}$ energy levels get close to each other during variation in Δ and will show singularity when the energy gap between them is closed at certain points Δ. As a result, the corresponding ***k***-integral $e_{222}^{\sigma (\pi )}$ would show critical behavior near this point.

To trace the possible singularity in the PBC, in Figure 2b, we plotted the energy gap between the lowest unoccupied and highest occupied levels for the σ-bands and π-bands; i.e., $E_{g}^{\sigma }=\underset{k,m,n}{\min}|E_{m\neq 8}^{0}(k)-E_{n\neq 7}^{0}(k)|$ and $E_{g}^{\pi }=\underset{k}{\min}|E_{8}^{0}(k)-E_{7}^{0}(k)|$, respectively. It can be seen that the $E_{g}^{\sigma }$ for σ-bands retained a finite value over the given interval in the Δ. Therefore, any Hamiltonian $H^{\sigma }(\Delta )$ lying in this range was adiabatically connectable to the non-piezoelectric $H^{\sigma }(0)$ without the appearance of singularity in $\Omega_{2,22}^{\sigma }$. Thereby, $e_{222}^{\sigma }$ should vary smoothly through zero with Δ. Noting that $E_{g}^{\sigma }\gg E_{g}^{\pi }$ and hence $\left|\Omega_{2,22}^{\sigma }|\ll |\Omega_{2,22}^{\pi }\right|$ over a quite large range of Δ, it is reasonable that $e_{222}^{\sigma }$ would have a relatively minor magnitude compared to the $e_{222}^{\pi }$ shown in Figure 2a.

However, the energy gap $E_{g}^{\pi }=|\Delta |$ for π-bands would close at Δ=0, across which a QPT occurs between quantum valley Hall states marked by the opposite valley Chern numbers $C_{v}=\mathrm{sgn}(\Delta )$ [12]. When Δ approaches this critical point, the PBC (Appendix A)
(9)$$\Omega_{2,22}^{\pi }(k)=C_{v}\frac{\beta_{pp\pi }at^{2}|\Delta |}{4{\left(E_{8}^{0}\right)}^{3}}\left[\cos\left(k\cdot \frac{\delta_{2}-\delta_{3}}{2}\right)\cos\left(k\cdot \frac{\delta_{3}-\delta_{1}}{2}\right)\cos\left(k\cdot \frac{\delta_{1}-\delta_{2}}{2}\right)-1\right]$$
and the MBC [29]
(10)$$\Omega_{\pi }(k)=C_{v}\frac{-\sqrt{3}a^{2}t^{2}|\Delta |}{2{\left(E_{8}^{0}\right)}^{3}}\left[\sin\left(k\cdot \frac{\delta_{2}-\delta_{3}}{2}\right)\sin\left(k\cdot \frac{\delta_{3}-\delta_{1}}{2}\right)\sin\left(k\cdot \frac{\delta_{1}-\delta_{2}}{2}\right)\right]$$
of the π-valence band will diverge. The trigonometry terms in the above square brackets are finite (∼1) for any ***k***, while at the K or $K^{\prime }$ points $k_{D}=(\pm 4\pi /3\sqrt{3}a,0)$, the energy of the π-conduction band in the denominators is $E_{8}^{0}=\sqrt{|f(k_{D}){|}^{2}+{\left(\Delta /2\right)}^{2}}=|\Delta |/2$. Thus, in the vicinity of $k_{D}$ where |f(k)|≪|Δ|, $\Omega_{2,22}^{\pi }$ and $\Omega_{\pi }$ will diverge as $\propto C_{v}\cdot \Delta^{-2}$ for a decreasing |Δ|, and then sharply peak around $k_{D}$ like the Dirac function, as shown in Figure 2c. For an sp^2^ crystal with TRS, $\Omega_{2,22}^{\pi }[\Omega_{\pi }]$ is an even [odd] function of ***k*** and satisfies $\Omega_{2,22}^{\pi }[\Omega_{\pi }](-\Delta )=-\Omega_{2,22}^{\pi }[\Omega_{\pi }](\Delta )$. In the |Δ|→0 limit, the even symmetric and Dirac-function-like distribution of $\Omega_{2,22}^{\pi }$ ensures that its ***k***-integral is non-vanishing, while $\Omega_{2,22}^{\pi }(-\Delta )=-\Omega_{2,22}^{\pi }(\Delta )$ requires that $e_{222}^{\pi }(0^{-})=-e_{222}^{\pi }(0^{+})$, thus leading to the abrupt sign-changing of $e_{222}^{\pi }$ when Δ varies from $0^{-}$ to $0^{+}$. Therefore, it was the singularity in $\Omega_{2,22}^{\pi }$ near the QPT point that gave rise to the critical behavior of $e_{222}^{\pi }$ shown in Figure 2a.

On the other hand, because $\Omega_{\pi }(-k)=-\Omega_{\pi }(k)$, its usual ***k***-integral given by the TKNN formula always vanishes and cannot show any criticality near the critical point at Δ=0. To capture the non-analyticity in the ground-state geometry of π-electrons during the QPT, an alternative way is to define an auxiliary quantity called the valley Hall conductance (VHC) [44]:(11)$$\sigma_{\mathrm{VH}}^{\pi }=\frac{e^{2}}{2\pi h}\left[\int_{K}dk\Omega_{\pi }(k)-\int_{K^{\prime }}dk\Omega_{\pi }(k)\right]$$
where h=2πℏ, and the subscripts K and $K^{\prime }$ denote the triangular domain in Figure 1b as delimited by the red *Γ*-*M* lines (along which $\Omega_{\pi }(k)=0$). It should be noted that in a TB model for π-bands, $\sigma_{\mathrm{VH}}^{\pi }$ is not quantized for any finite non-zero Δ [44]. As illustrated in Figure 2d, $\sigma_{\mathrm{VH}}^{\pi }$ showed quite similar variation trends to those of $e_{222}^{\pi }$. The discontinuity of $\sigma_{\mathrm{VH}}^{\pi }$ at the QPT point can also be explained by essentially the same argument employed for that of $e_{222}^{\pi }$. This similarity between the critical behaviors of $\sigma_{\mathrm{VH}}^{\pi }$ and $e_{222}^{\pi }$ during the QPT motivated us to further explore their intimate relations in the following.

### 3.2. Valley Model for the Anomalous π-Piezoelectricity

Having discerned the dominant role of π-electrons in the piezoelectricity of sp^2^ crystals, we now turn to closely examining the microscopic mechanism underneath the anomalous π-piezoelectricity and its subtle relation to the valley Hall effect as suggested by the TB calculation.

#### 3.2.1. Correlation between π-Piezoelectric Coefficient and VHC

Owning to the sharply peaked distribution of the PBC in the K and $K^{\prime }$ valleys, the main physics of the π-piezoelectricity in sp^2^ crystals can be captured by an analytical valley model based on the low-energy effective Hamiltonians. When expanding $H^{\pi }(k,\varepsilon )$ in Equation (4) at $k_{D}=(\pm 4\pi /3\sqrt{3}a,0)$, we obtain the massive Dirac Hamiltonian:(12)$$H^{\tau }(q,\varepsilon )=\sigma_{3}\Delta /2+\hbar v_{F}[\tau \sigma_{1}(q_{1}-\tau {\tilde{A}}_{1})+\sigma_{2}(q_{2}-\tau {\tilde{A}}_{2})]$$
where $\hbar v_{F}=3at/2$ is the Fermi velocity; $\sigma_{1}$, $\sigma_{2}$, and $\sigma_{3}$ are the Pauli matrices; τ=± refers to the K or $K^{\prime }$ valley (when appearing in the sub- or superscript of a variable τ=K or $K^{\prime }$); and $q=(q_{1},q_{2})$ is the crystal momentum measured from the $k_{D}$ point. The strain ε emerges as a pseudo-gauge field given by ${\tilde{A}}_{i}=\beta_{pp\pi }/2a\sum_{jk}\gamma_{ijk}\varepsilon_{jk}$; here, the non-zero elements of the rank-3 tensor $\gamma_{ijk}$ are restricted by the D_3h_ symmetry to obey $-\gamma_{111}=\gamma_{122}=\gamma_{212}=\gamma_{221}=1$ [31]. In real sp^2^ crystals, the lattice constant a offers a natural high-energy cutoff of q=|q|≤Λ∼1/a for the above Hamiltonian, beyond which the Dirac approximation is inapplicable [30].

By starting from $H^{\tau }(q,\varepsilon )$, repeating the derivation procedure of Equation (9), and exploiting the linear mapping relationship $\partial_{\varepsilon_{jk}}\to \partial_{q_{i}}$, we find that the PBC of the τ-valley valence band is (see Appendix A):(13)$$\Omega_{i,jk}^{\tau }(q)=-\sum_{l}\tau \frac{\epsilon_{il}\gamma_{ljk}\beta_{pp\pi }}{2a}\Omega_{\tau }(q),$$
where $\Omega_{\tau }(q)=\tau 9a^{2}t^{2}\Delta /2{\left[{\left(3atq\right)}^{2}+\Delta^{2}\right]}^{3/2}$ is just the MBC for the τ-valley valence band [32], and $\epsilon_{il}$ is the antisymmetric Levi–Civita tensor. Then, by adding up the integrals of $\Omega_{i,jk}^{\tau }(q)$ over the K and $K^{\prime }$ valleys, the π-piezoelectric tensor can be obtained as:(14)$$e_{ijk}^{v}=2e\sum_{\tau }\int_{q\leq \Lambda }\frac{dq}{{\left(2\pi \right)}^{2}}\Omega_{i,jk}^{\tau }(q)=-\sum_{l}\frac{\epsilon_{il}\gamma_{ljk}\hbar \beta_{pp\pi }}{ea}\sigma_{\mathrm{VH}}^{v}$$

Thus, in the valley model (labeled with the superscript “v”), $e_{ijk}^{v}$ is directly related to the VHC $\sigma_{\mathrm{VH}}^{v}=\sigma_{H}^{K}-\sigma_{H}^{K^{\prime }}$ with $\sigma_{H}^{\tau }=(e^{2}/h)\int_{q\leq \Lambda }dq\Omega_{\tau }(q)/2\pi$. When integrating out the crystal momentum q straightforwardly, we arrive at:(15)$$\sigma_{\mathrm{VH}}^{v}=\frac{e^{2}}{h}\left[1-\frac{1}{\sqrt{9{\left(a\Lambda /\alpha \right)}^{2}+1}}\right]C_{v},$$
where α=|Δ/t| measures the polarity of π-bonds [24], and the energy cutoff is set as $\Lambda =\sqrt{S_{BZ}/2\pi }$ to preserve the total number of states in the first BZ with an area of $S_{BZ}=8\pi^{2}/3\sqrt{3}a^{2}$ [30]. When substituting Equation (15) into Equation (14), we finally obtain the explicit expression for the piezoelectric coefficient:(16)$$e_{ijk}^{v}=-\sum_{l}\epsilon_{il}\gamma_{ljk}\frac{e\beta_{pp\pi }}{2\pi a}\left[1-\frac{1}{\sqrt{4\sqrt{3}\pi /\alpha^{2}+1}}\right]C_{v}.$$

Here, the factor $-\sum_{l}\epsilon_{il}\gamma_{ljk}$ ensures that the components of $e_{ijk}^{v}$ satisfy the D_3h_ symmetry; i.e., $e_{211}^{v}=e_{112}^{v}=e_{121}^{v}=-e_{222}^{v}$ and $e_{111}^{v}=e_{122}^{v}=e_{212}^{v}=e_{221}^{v}=0$ [15]. Roughly, $\beta_{pp\pi }∼l_{p}+{l^{\prime }}_{p}+1$ is determined by the angular momentum of the involved $p_{z}$ orbitals $l_{p}={l^{\prime }}_{p}=1$ [42] and therefore can be treated as constant in the following.

We will now discuss the mechanisms of the π-piezoelectricity revealed by Equations (14)–(16). Firstly, $e_{ijk}^{v}$ is an inverse proportional function of the bond length a, and thus it tends to diminish with an elongating a; secondly, as inherited from $\sigma_{\mathrm{VH}}^{v}$, $e_{ijk}^{v}$ is also a decreasing function of the bond polarity α. Hereinafter, these two trends will be referred to as “the bond length and polarity mechanisms”, respectively. When α≪1, the VHC is nearly quantized as $\sigma_{\mathrm{VH}}^{v}\approx C_{v}\cdot e^{2}/h$, and thus the π-piezoelectric coefficient can be approximated as $e_{222}^{v}\approx C_{v}\cdot e\beta_{pp\pi }/2\pi a$. However, this α-independent formula becomes physically unreasonable and would overvalue the magnitude of $e_{222}^{v}$ for partially ionic sp^2^ crystals such as h-BN (α>2, see Table 2) [21]. Indeed, recent ab initio studies showed that the piezoelectricity in ISB graphene would decay significantly with an increasing effective α (or Δ) [25].

The valley Chern number $C_{v}$ in Equation (16) reflects the topological aspects of the π-piezoelectricity in sp^2^ crystals. Although $\sigma_{\mathrm{VH}}^{v}$ itself is not quantized for a finite Δ≠0, its jump of $N_{3}=h[\sigma_{\mathrm{VH}}^{v}(0^{+})-\sigma_{\mathrm{VH}}^{v}(0^{-})]/e^{2}=2C_{v}(\Delta >0)$ when crossing the critical point at Δ=0 is exactly quantized. Actually, as pointed out in [11], $N_{3}$ is a well-defined topological invariant, and in the spirit of the bulk-edge correspondence, was related to the number of kink states or the zero-energy modes [45] that existed at the interface between the two pieces of sp^2^ crystals with an opposite sgn(Δ) shown in Figure 3. In this sense, the QPT between ground states of different $C_{v}$ is regarded as marginally topological [46]. In sync with $\sigma_{\mathrm{VH}}^{v}$, the jump of $e_{222}^{v}$ across Δ=0 should also be quantized (in units of $e\beta_{pp\pi }/\pi a$) according to Equation (14) and thereby can serve as an alternative signal for probing such topological QPT in sp^2^ crystals [12,14].

#### 3.2.2. π-Piezoelectricity as a Hall-Type Response to Pseudo-Electric-Field

From a dynamic viewpoint, the anomalous π-piezoelectric response can be intuitively illustrated by drawing an analogy with the valley Hall effect. Consider now that the strain ε(t) is adiabatically time-dependent. The resultant pseudo-gauge potential that couples to the τ-valley electrons through the Peierls substitution $q\to q-\tau \tilde{A}(t)$ will act as an electric field ${\tilde{E}}^{\tau }=-\tau \dot{\tilde{A}}(t)$ with opposite directions for non-equivalent valleys [47]. This pseudo-electric field is distinct from the real electric field $E=-\dot{A}(t)$ yielded by the varying magnetic potential A(t), which couples to both valleys with the same signs. It is well known [32] that in the valley Hall effect illustrated in Figure 4a, a longitudinally applied real electric field E will drive the valley-contrasting transverse Hall current of $J^{\tau }(t)={\hat{\sigma }}_{H}^{\tau }\times E$ (here, ${\hat{\sigma }}_{H}^{\tau }=\sigma_{H}^{\tau }\hat{z}=\pm \sigma_{\mathrm{VH}}^{v}\hat{z}/2=\pm {\hat{\sigma }}_{\mathrm{VH}}^{v}/2$). Since $J^{K}=-J^{K^{\prime }}$, the charge current $J^{K}+J^{K^{\prime }}=0$ and hence no Hall voltage will be measured, albeit one can define a valley current $J^{v}=J^{K}-J^{K^{\prime }}={\hat{\sigma }}_{\mathrm{VH}}^{v}\times E$ [44]. On the other hand, the piezoelectric current contributed by the τ-valley takes the following form (Appendix A):(17)$${\tilde{J}}^{\tau }=(2\hbar /e){\hat{\sigma }}_{H}^{\tau }\times {\tilde{E}}^{\tau }.$$
which allows an intuitive reinterpretation of ${\tilde{J}}^{\tau }$ as a Hall-type current driven by the pseudo-electric field ${\tilde{E}}^{\tau }$. Differing from the canceling-out by $J^{\tau }$ in the charge-neutral valley Hall effect, the strain-induced ${\tilde{J}}^{\tau }$ can add up to result in a non-zero charge current ${\tilde{J}}^{v}={\tilde{J}}^{K}+{\tilde{J}}^{K^{\prime }}$ $=(2\hbar /e)\cdot {\hat{\sigma }}_{\mathrm{VH}}^{v}\times \dot{\tilde{A}}$ due to ${\tilde{E}}^{K}=-{\tilde{E}}^{K^{\prime }}$. Its accumulation during t∈[0,T] yields a bulk polarization of
(18)$$\Delta P^{v}=\int_{0}^{T}dt{\tilde{J}}^{v}(t)=(2\hbar /e)\cdot {\hat{\sigma }}_{\mathrm{VH}}^{v}\times \tilde{A}(\varepsilon (T))$$
and a potential difference between two edges of the sample shown in Figure 4b, which in an analogous sense can be regarded as the “Hall voltage”. To summarize, in a dynamic picture, the anomalous π-piezoelectric effect is a charge-non-neutral counterpart of the valley Hall effect [48]. Such an immediate correlation between them may have important experimental implications for measuring the VHC of sp^2^ systems via the piezoelectric effect.

### 3.3. Application to Typical sp^2^ Crystal and BNG Superlattice

Given the π-electrons’ predominant contribution to the total piezoelectricity compared to that of the σ-electrons, the central concern in this section is the extent to which the piezoelectricity of real sp^2^ materials can be determined by their π-electrons. Below, we first apply our π-band models to the typical sp^2^ crystals, including h-SiC and BX (X = N, P, As, Sb), and then to the BNG superlattice with D_3h_ symmetry to evaluate their piezoelectricity.

#### 3.3.1. Intrinsic Piezoelectricity of Typical Sp^2^ Crystals

Let us calculate the π-piezoelectric coefficients of the typical sp^2^ crystals including h-SiC and BX using our established models. The TB parameters for the π-bands of these unstrained crystals can be evaluated by fitting their DFT band structures (see Appendix A). In Table 2, we first list the bond length a and the fitted t and Δ. Then, we calculated their bond polarity α and VHC $\sigma_{\mathrm{VH}}^{v}$. By adopting a unified constant $\beta_{pp\pi }=3.3$ that was well fitted from graphene and h-BN [20,31], we further evaluated their π-piezoelectric coefficients $e_{222}^{\pi (v)}$ using the TB (valley) model and compared the obtained $e_{222}^{\pi (v)}$ with the clamped-ion DFT results $e_{222}$ (corresponding to our models [5]) in the last three columns. To avoid drawing biased conclusions due to the underestimation of band gaps (Δ) in the DFT, the GW-corrected TB parameters cited from the extant literature, together with the data calculated from them, are also listed as comparisons in parentheses. Since the GW gaps were systematically larger than the DFT gaps, the values of $e_{222}^{\pi (v)}$ calculated from the GW data were smaller than those from the DFT data. Despite this, we safely concluded from Table 2 that for all these crystals, the π-band contribution $e_{222}^{\pi (v)}$ accounted for a major part of the total electronic piezoelectric coefficient $e_{222}$, while the remaining proportion attributed to the σ-bands had a relatively minor contribution. For instance, the ratio of $e_{222}^{\pi (v)}/e_{222}$ for the partially ionic BN was about 80% (70%), which confirmed the previous DFT calculation results [43]; for the nearly covalent BP and BAs, this ratio could be even higher (up to 95% (85%)), meaning that the piezoelectricity was almost entirely determined by the π-bands.

Moreover, we noticed that as the crystal changed from BN to BSb along the first column of Table 2, the bond polarity α decayed monotonically and was accompanied by an elongation in the bond length a. According to the bond polarity mechanism in Equation (16), the decline in α tended to enhance the π-piezoelectricity via increasing $\sigma_{H}^{v}(\alpha )$; meanwhile, the concomitant elongation of a restrained this enhancement via the bond-length mechanism. As illustrated in Figure 5, owing to the rapid decay in α, the increase in $\sigma_{\mathrm{VH}}^{v}(\alpha )$ and $e_{222}^{v}$ when going from BN to BP was dominated by the bond polarity mechanism. Thereafter, since $\sigma_{H}^{v}(\alpha )$ gradually saturated, the once-hidden bond-length mechanism came into play through the factor 1/a and partially canceled the feeble growth of $\sigma_{H}^{v}(\alpha )$, thus leading to the plateau or slight decline in $e_{222}^{v}$ for the subsequent BAs and BSb. Therefore, the π-piezoelectricity in these sp^2^ crystals was determined by the competition or balance between the above two mechanisms. This in turn would largely govern their overall piezoelectricity considering the subordinate role of σ-electrons. For example, the obvious difference in $e_{222}$ between the partially ionic BN and SiC (α>1) and the nearly covalent BP, BAs, and BSb (α<1) could be roughly attributed to the apparent bond polarity difference between these two groups.

#### 3.3.2. Engineered Piezoelectricity of BNG Superlattice

Apart from the perfect sp^2^ crystals, many engineered sp^2^ systems such as graphene on substrates [34], F and H absorbed graphene [35], and BNG [36] can also be piezoelectric. Among these, the BNG superlattice seems to be the most promising platform for piezoelectric engineering due to its experimentally easily tailorable electronic structures [55]. To validate our models in guiding the engineering of piezoelectricity in sp^2^ systems, we took as an example the D_3h_-symmetric BNG superlattice shown in Figure 5a, the piezoelectricity of which was first studied using a DFT calculation in [36]. Its primitive vector was p(≥2) times as long as that of the pristine graphene. Accordingly, as shown in Figure 5b, the pristine BZ was folded into $1/p^{2}$ of the original one to form a superlattice BZ. The periodically embedded (BN)_3_ rings broke the inversion symmetry and opened a band gap of graphene by introducing sublattice-dependent perturbations into it (since the B and N atoms resided in different sublattices). This made the BNG superlattice piezoelectric.

According to the band-folding picture [56], the low-energy-band structures of BNG superlattices fall into two categories depending on whether p is a multiple of 3. In the p≠3n (n is an integer) scenarios in which (BN)_3_-mediated scattering between K and $K^{\prime }$ valleys is suppressed, the BNG superlattice can be approximately treated as ISB graphene crystal within the virtual crystal approximation (VCA) [57,58,59]: both the on-site and hopping energies are averaged separately over each sublattice of the BNG to form a virtual sp^2^ crystal; thus, the low-energy dispersion of the deformed BNG could still be described by the effective Hamiltonian in Equation (12) except that the parameters t(x) and Δ(x) depended on the BN concentration $x=3/p^{2}$. Predictably, their piezoelectricity could be sufficiently evaluated using our valley model. However, when p=3n, the K and $K^{\prime }$ points of the primitive BZ (red hexagon in Figure 6b) were folded onto the Γ points of the superlattice BZ [56], near which the two valleys were degenerate and the scattering between them was strongly enhanced, thus invalidating the VCA. In this situation, minimally describing the degenerate valleys would require a 4×4 valley-coupled Hamiltonian [60]; exploring the piezoelectricity in such systems entails a non-abelian model [7], which was beyond the scope of the current work and therefore was left for future study.

In the following, we will focus only on the BNG superlattices with p≠3n configurations. We used Equation (16) at the mean-field VCA level to estimate their π-piezoelectric coefficients from their corresponding virtual sp^2^ crystals. Since $E_{g}^{\pi }=|\Delta |$, the effective ISB strength of the virtual crystals could be derived from the DFT band gaps of the BNG superlattices: |Δ(x)|=4.11x(eV) [36]. If we assumed that their effective hopping varied linearly with *x* from −2.30 eV for h-BN [49] to −2.80 eV for graphene [39]; i.e., t(x)=−2.80+0.50x(eV), then the effective bond polarity was given by $\alpha (p)=12.33/(2.80p^{2}-1.50)$. Inserting this into Equation (16), we obtained:(19)$$e_{222}^{v}(p)=\frac{e\beta_{pp\pi }}{2\pi a}\left(1-\frac{12.33}{\sqrt{4\sqrt{3}\pi {\left(2.80p^{2}-1.50\right)}^{2}+12.33^{2}}}\right)C_{v}$$

In Figure 7, the calculated $e_{222}^{v}(p)$ when adopting a=1.42 Å and $\beta_{pp\pi }=3.3$ for all of the considered BNG superlattice configurations (Δ>0) were compared with the clamped-ion DFT results ($e_{222}(p)$). It can be seen that our valley model, albeit only when including the π-band contributions, provided an accurate-enough estimation for the $e_{222}(p)$ of the BNG superlattices, especially in the large p (small *x*) cases. This was not surprising when considering that as the virtual sp^2^ crystal’s α(p) decayed with the enlarging p, its σ-piezoelectricity became gradually neglectable, so that $e_{222}(p)\approx e_{222}^{v}(p)$. Since the bond length was fixed, $e_{222}^{v}(p)$ increased with the decaying α(p) solely through the bond polarity mechanism. In the p→∞ limit, it saturated at an ultrahigh value of $e_{222}^{v}(\infty )=e\beta_{pp\pi }/2\pi a=5.88\times$$10^{-10}\;C/m$, which agreed well with the DFT result $e_{222}(\infty )=5.50\times 10^{-10}\;C/m$ [36]. Such asymptotic piezoelectricity, in corresponding to a quantized VHC $\sigma_{\mathrm{VH}}^{v}(\infty )=e^{2}/h,(C_{v}=1)$, was protected by the marginal valley topology [46] (see also Section 3.2.1), and hence seemed robust against the change of ISB inducers. For example, the further DFT study in [36] showed that after replacing the (BN)_3_ rings in BNG superlattices with D_3h_ hole defects, the calculated value for $e_{222}(\infty )$ was found to be about $5.60\pm 0.4\times 10^{-10}\;C/m$, which was almost unchanged compared to its original value.

Arguably, even when the ion relaxation was considered, this π-electron-governed and topologically robust piezoelectricity of the BNG superlattice was still expected. Within the rigid-ion picture, the ionic piezoelectricity of the relaxed BNG superlattice mainly arose from the strain-induced dipole change in the (BN)_3_ clusters (because the iso-charged C^4+^ ions contributed no net dipole), which, when averaged over the supercell, would decrease with the enlarging *p* just like the σ-piezoelectricity. Therefore, in the p→∞ limit, the ion-relaxation correction to the piezoelectricity primarily affected the π-electrons’ response, which was embodied as the renormalization of the electron–lattice coupling parameter $\beta_{pp\pi }$ into $\beta_{pp\pi }^{rel}$. When adopting an accurate value of $\beta_{pp\pi }^{rel}=2.66$ for relaxed-ion graphene [61], the asymptotic piezoelectric coefficient for the relaxed BNG superlattice was calculated as $e_{222}^{v-rel}(\infty )=4.74\times 10^{-10}\;C/m$, which again agreed well with the DFT result of $e_{222}^{rel}(\infty )=4.50\;\times$ $10^{-10}\;C/m$ [36]. This experimentally measurable piezoelectric coefficient was about one order larger than that of the F and H absorbed graphene C_2_HF ($e_{222}^{rel}=0.63\times 10^{-10}$
C/m) [35] and three times larger than that of the h-BN monolayer ($e_{222}^{rel}=1.38\times 10^{-10}\;C/m$) [15]. Based on the above discussion, we concluded that, as in the cases of perfect sp^2^ crystals, the piezoelectricity in the BNG superlattice was also controlled to a large extent by the π-electrons.

We will now discuss the feasibility of π-piezoelectric engineering in real sp^2^ materials and propose some suggestions for this purpose based on our results. Compared with strong σ-bonds, low-energy π-bonds are more sensitive to structural, physical, or chemical modifications [34,35,36,55]. These external perturbations will inevitably reshape the ground-state geometry of π-electrons and thereby affect their piezoelectric response. Such a tunability of π-electrons combined with their decisive role in piezoelectricity makes the efficient engineering of piezoelectricity in sp^2^ materials possible by predesigning their π-band structures. Because the anomalous piezoelectric response of π-bands stems from its non-trivial valley geometry (as reflected by the non-zero VHC), to maximally exploit the latent π-piezoelectricity, the first suggestion is that the engineering schemes should be designed to avoid destroying the valley structures. This in turn may explain why C_2_HF has a one-order-smaller piezoelectric coefficient than that of the BNG [35]: the absorbent F and H atoms fix all the non-local π-electrons of graphene into strong C-F and C-H σ-bonds and thereby push their energy far away from the Fermi-level, even beyond the C-C σ-bands [62], and then consequently destroy the low-energy valleys. Upon reserving the valley structure of π-bands, higher π-piezoelectricity would benefit from a shorter bond length or weaker bond polarity in the VCA sense. With the shortest bond lengths in the sp^2^ family, graphene (a=1.42 Å) and h-BN (a=1.44 Å) seem to be the ideal starting crystals for π-piezoelectric engineering, as suggested by the bond-length mechanisms. Once the starting crystal is selected, external modifications generally cause hardly any dramatic change in its bond length, and thus manipulate its π-piezoelectricity mainly via the bond-polarity mechanisms. Therefore, another suggestion based on this consideration is that the π-piezoelectricity hidden in the starting crystals can be further released by engineering strategies that lower its effective VCA bond polarity (or equally narrow the band gap |Δ| relative to t). For example, after hybridizing with carbon to form borocarbonitrides (B_x_C_y_N_x_) [23], the band gap of the h-BN monolayer was dramatically narrowed, and as a result its relaxed-ion piezoelectric coefficient could be significantly enhanced from the original $e_{222}^{rel}=1.38\times 10^{-10}\;C/m$ up to $e_{222}^{rel}=5.00\times 10^{-10}\;C/m$ for the BC_2_N structures.

Before closing this section, we will briefly discuss how the piezoelectricity in sp^2^ monolayers can be observed. For crystals with a moderate band gap such as h-SiC, the piezoelectricity is usually observed via a transport measurement [16] in which the oscillating piezoelectric current is detected as a response to the cyclic deformation. However, this method is not suitable for sp^2^ monolayers such as h-BN, the large band gap of which impedes the measurement of current. As an alternative, we highly recommend an innovative approach recently proposed in [63]; i.e., observing the piezoelectric effect by detecting the piezo charges induced by inhomogeneous in-plane deformations ε(r). In this scheme, the density of piezo charges is given by $\rho_{piezo}=-\nabla_{r}\cdot P=e_{222}[2\partial_{x}\varepsilon_{xy}+\partial_{y}(\varepsilon_{xx}-\varepsilon_{yy})]$. For a given strain configuration, $e_{222}$ can be inferred from the distribution of $\rho_{piezo}$ measured using electrostatic force microscopy (EFM) [19]. For example, in the simplest situation in which the lattice deformation has a constant strain gradient *c* along the *y* direction $\varepsilon (r)=(0,cy+c_{0})$, the piezoelectric coefficient can be easily determined from the uniform piezo charge density as $e_{222}=-\rho_{piezo}/c$.

In real EFM experiments, due to the in-plane polarizability of sp^2^ monolayers [64], the piezoelectric field will be further screened, thus leading to the partial compensation of $\rho_{piezo}$. In other words, the total charge density under inhomogeneous strains should contain the piezo- and screening-induced parts: $\rho =\rho_{piezo}+\rho_{ind}$. The latter is determined by $\rho_{ind}=\chi_{2D}\nabla_{r}^{2}\varphi (r)$, where φ(r) is the piezo-charge-induced electric potential, and $\chi_{2D}$ is the 2D polarizability of the sp^2^ monolayers [63]. Using the Poisson equation $\nabla_{r}^{2}\varphi =-4\pi \rho$, the piezo charge density is related to the piezo potential via $\rho_{piezo}=-(\chi_{2D}+1/4\pi )\nabla_{r}^{2}\varphi$. In the simplest example considered above, one can further extract $e_{222}$ from the measured φ(y) as $e_{222}=(\chi_{2D}+1/4\pi )\partial_{y}^{2}\varphi (y)/c$. Since the observable piezo potential φ(y) is reduced by the compensation charges $\rho_{ind}$, ignoring this screening effect will lead to an incorrectly extracted piezoelectric coefficient $e_{222}^{*}=\partial_{y}^{2}\varphi (y)/4\pi c$. Therefore, when observing the piezoelectricity of sp^2^ monolayers via the EFM measurement, their in-plane polarizability must be carefully treated.

## 4. Conclusions

In summary, in this work, the bond-orbital-resolved electronic piezoelectricity in sp^2^-hybridized monolayer semiconductors was systematically investigated by combining the TB method and the geometric phase theory of polarization. We revealed in the TB calculation that their π- and σ-piezoelectricity showed contrasting variations trends with the ISB strength Δ. Unlike the continuous decrease in the subordinate σ-piezoelectricity with a decaying |Δ|, the predominant π-piezoelectricity increased piecewise as |Δ|→0 and displayed critical behavior at the QPT point Δ=0 due to the non-analyticity of the π-electrons’ ground-state geometry near this point. By focusing on the anomalous piezoelectric response of π-electrons, we further related the π-piezoelectricity to the valley Hall effect and reinterpreted it as a Hall-type response to the strain-induced pseudo-electric field in the low-energy valley model. With the help of this analytical model, we also identified the bond-length and bond-polarity mechanisms that underlie π-piezoelectricity and clarified its topological aspects. The validity of our theoretical predictions that the piezoelectricity of these materials is mainly dominated by the anomalous response of π-electrons was quantitatively justified by applying the π-band models to the typical sp^2^ crystals h-SiC and BX, as well as the BNG superlattice. Our investigation not only deepens the understanding of piezoelectricity in real sp^2^ materials, but also provides guidelines for tailoring their piezoelectricity, thus opening doors to π-electron piezoelectric engineering in these systems.

## Figures and Tables

**Figure 1 materials-15-07788-f001:**
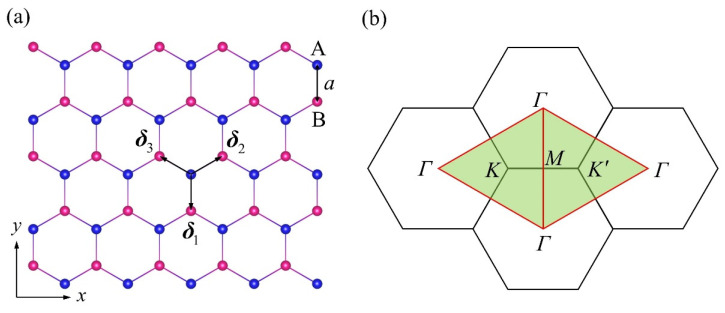
(**a**) Undeformed crystal structure of the sp^2^ piezoelectrics with three nearest-neighbor vectors $\delta_{I=1,2,3}$ of magnitude $a=|\delta_{I}|$ connecting its inequivalent atoms A (blue) and B (red). (**b**) The corresponding BZs, which are demarcated into two triangular subzones by the *Γ*-*M* mirror lines (red) for each valley centered at K and $K^{\prime }$ points.

**Figure 2 materials-15-07788-f002:**
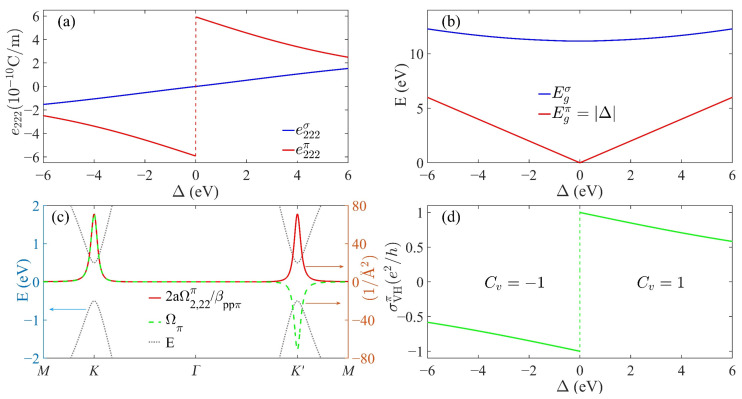
(**a**) Bond-orbital-resolved piezoelectric coefficients $e_{222}^{\sigma }$ (blue line) and $e_{222}^{\pi }$ (red line) as functions of the ISB strength Δ. (**b**) The energy gap between the highest occupied and lowest unoccupied levels $E_{g}^{\sigma }$ for σ-bands (blue line) and $E_{g}^{\pi }$ for π-bands (red line). (**c**) Distributions of $\Omega_{2,22}^{\pi }(k)$ (red solid line) and $\Omega_{\pi }(k)$ (green dashed line) of the π-valence bands (gray dotted line) along high symmetry ***k***-path in the BZ. The parameters used were t=−2.8eV and Δ=0.3eV. (**d**) The variation in the VHC $\sigma_{\mathrm{VH}}^{\pi }$ as a function of Δ.

**Figure 3 materials-15-07788-f003:**
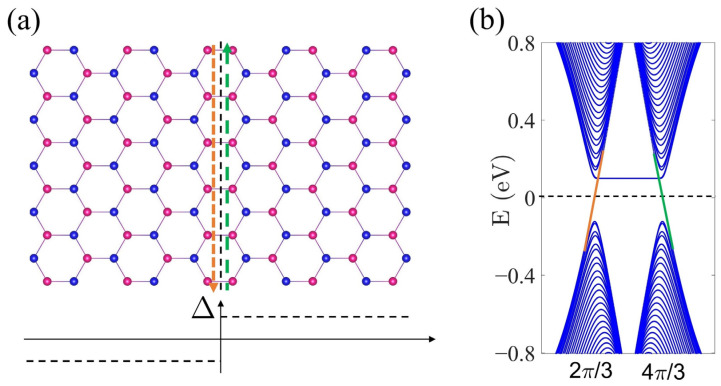
(**a**) Schematic diagram of the zigzag domain-wall (upper-panel) constructed by two pieces of sp^2^ crystals with Δ=±0.3 eV (lower panel). (**b**) The corresponding band structure for the π-electrons for t=−2.8 eV. The kink states situated at the zero-energy level in (**b**) and their real-space propagation direction along the domain-wall in (**a**) are marked in different colors.

**Figure 4 materials-15-07788-f004:**
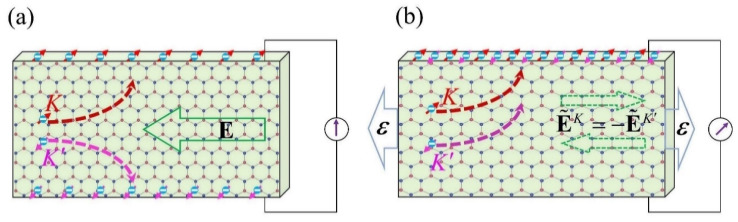
Schematic diagram illustrating the trajectories of valley electrons driven by (**a**) the real electric field and (**b**) strain-induced pseudo-electric field.

**Figure 5 materials-15-07788-f005:**
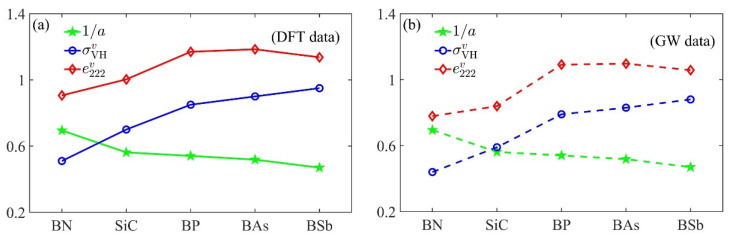
The variation trends in 1a(Å−1), $\sigma_{\mathrm{VH}}^{v}(e^{2}/h)$, and $e_{222}^{v}(\beta_{pp\pi }\times 10^{-10}\;C/m)$ as the material changed from BN to BSb. Panels (**a**) and (**b**) were plotted using the DFT and GW-corrected data, respectively.

**Figure 6 materials-15-07788-f006:**
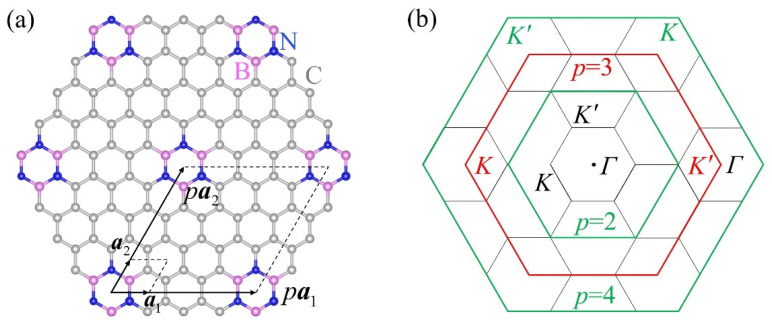
(**a**) Schematic diagram of the BNG superlattice with its p×p supercell (here, p=4) marked by the black rhombus. (**b**) Illustration of the BZ folding: the larger hexagons with a red (for p=3n) or green (for p=3n±1) edge are the primitive BZs relative to the fixed superlattice BZ (small black hexagon).

**Figure 7 materials-15-07788-f007:**
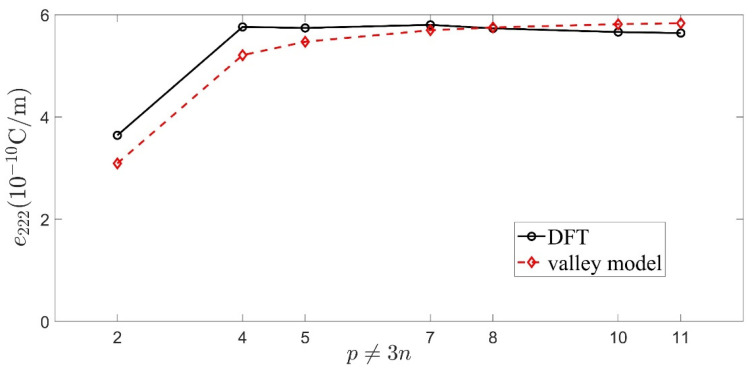
Comparison between $e_{222}^{v}(p)$ calculated using Equation (19) and the clamped-ion DFT results $e_{222}(p)$ [36] for the BNG superlattices with p≠3n configurations and Δ>0.

**Table 1 materials-15-07788-t001:** The strain-modified nearest-neighbor hopping $t_{\mu ,\mu^{\prime }}^{I}(\varepsilon )$ expressed in terms of Slater–Koster two-center integrals $V_{\chi }(\varepsilon )$ [37]. The direction cosines of vector pointed from the μ-orbital site to the $\mu^{\prime }$-orbital site is denoted by ${\hat{n}}_{\mu ,\mu^{\prime }}^{I}=(n_{x},n_{y},n_{z})$. Other elements can be found by permuting indices.

$t_{s,s}^{I}(\varepsilon )$	$V_{ss\sigma }(\varepsilon )$	$t_{p_{x},p_{x}}^{I}(\varepsilon )$	$n_{x}^{2}V_{pp\sigma }(\varepsilon )+(1-n_{x}^{2})V_{pp\pi }(\varepsilon )$
$t_{s,p_{x}}^{I}(\varepsilon )$	$n_{x}V_{sp\sigma }(\varepsilon )$	$t_{p_{x},p_{y}}^{I}(\varepsilon )$	$n_{x}n_{y}[V_{pp\sigma }(\varepsilon )-V_{pp\pi }(\varepsilon )]$
$t_{s,p_{z}}^{I}(\varepsilon )$	$n_{z}V_{sp\sigma }(\varepsilon )$	$t_{p_{x},p_{z}}^{I}(\varepsilon )$	$n_{x}n_{z}[V_{pp\sigma }(\varepsilon )-V_{pp\pi }(\varepsilon )]$

**Table 2 materials-15-07788-t002:** The input parameters a(Å), Δ(eV), t(eV), and α of h-SiC and BX used to calculate their $\sigma_{\mathrm{VH}}^{v}(e^{2}/h)$ and $e_{222}^{\pi (v)}$. The calculated $e_{222}^{\pi (v)}$ values are compared with the clamped-ion DFT results for $e_{222}$ in units of $10^{-10}C/m$. The results corresponding to the GW band structures are given in parentheses.

Materials	a	Δ	t	α	$\sigma_{\mathrm{VH}}^{v}$	$e_{222}^{\pi }$	$e_{222}^{v}$	$e_{222}$
BN	1.44 ^a^	6.00 ^a^ (7.25 ^b^)	−2.30 ^a^ (−2.30 ^b^)	2.61 (3.15)	0.51 (0.44)	2.65 (2.17)	2.99 (2.57)	3.71 ^c^
SiC	1.78 ^d^	2.56 ^d^ (3.48 ^e^)	−1.74 ^d^ (−1.64 ^f^)	1.47 (2.12)	0.70 (0.59)	3.26 (2.57)	3.31 (2.77)	3.70 ^g^
BP	1.85 ^d^	1.30 ^d^ (1.83 ^h^)	−1.82 ^d^ (−1.84 ^i^)	0.71 (0.99)	0.85 (0.79)	4.00 (3.69)	3.86 (3.60)	4.25 ^j^
BAs	1.93 ^d^	0.70 ^d^ (1.24 ^k^)	−1.46 ^d^ (−1.55 ^l^)	0.48 (0.80)	0.90 (0.83)	4.02 (3.72)	3.91 (3.62)	4.32 ^j^
BSb	2.13 ^d^	0.33 ^d^ (0.73 ^h^)	−1.40 ^d^ (−1.34 ^l^)	0.24 (0.55)	0.95 (0.88)	3.86 (3.62)	3.75 (3.49)	4.51 ^j^

^a^ [20], ^b^ [49], ^c^ [15], ^d^ (Appendix A), ^e^ [50], ^f^ [51], ^g^ [26], ^h^ [52], ^i^ [53], ^j^ [22], ^k^ [28], ^l^ [54].

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
