# Peer review of "Bond-Orbital-Resolved Piezoelectricity in Sp2-Hybridized Monolayer Semiconductors"

_materials, 2022, doi:10.3390/ma15217788_

Round 1
Reviewer 1 Report
The proposed theoretical modelling of piezoelectric parameters of honeycomb crystals is not the first attempt to develop semi-analytical theory for the effect, and as a continuation of the earlier effort it would require paying more attention to the quantitative validity of the used approximations. In this respect, I find that the presented calculations have two flaws:
a) DFT modelling used is not done with the full GW methodology, hence, uses the underestimated gaps between sublattice-polarised bands in, e.g., hBN.
b) the calculation also lacks analysis of in-plane polarisability of 2D materials (see e.g., in PRL 114, 107401 (2015)), potentially, leading to a partial compensation of charges assigned to the lattice deformations and certainly reducing the observable piezopotentials.
c) it would be useful to complete the text with examples for how the piezoeffects can be observed (for examples, see in PHYSICAL REVIEW LETTERS 124, 206101 (2020)).
Reviewer 2 Report
Authors of this work have systematically examined the bond-orbital-resolved electronic piezoelectricity (i.e., the π- or σ-piezoelectricity) of Sp2-hybridized monolayer semiconductors using tight-binding piezoelectric model. They have also applied the model to the typical sp2-crystals including hexagonal silicon carbide and Boron-X (X=N, P, As, Ab), as well as the BN-doped graphene superlattices, to evaluate their intrinsic or engineered piezoe-lectricity. The results discussed are fundamental by nature. However, there are a few things that should be corrected during a revision.
1. The abstract of the paper does not reveal what are these “Sp2-hybridized monolayer semiconductors”? Moreover, the entire abstract must be rewritten with clarify since it is too much tedious in its current form.
2. The introduction of the paper must be reduced to a maximum of one page.
3. Details of DFT calculations involved are not clarified in the Methods and Formulas section
4. The length of the paper is too long, and it should be reduced significantly. I believe that this is due to several unnecessary (standard) equations that appeared all around the paper, which should be provided as supplementary.
5. Figure 2 is too blurry, and I cannot read the fonts without enlarging the PDF file to 200%
6. There are several typos and grammatical errors that should be corrected.
Round 2
Reviewer 2 Report
I repeat that authors of the work did not grasp my previous comments. In fact, this paper is based on the results of DFT calculations, among others. In their reply, authors wrote that DFT-based calculation details are not necessary in the methods and formula section. This puzzles me greatly. Rather, they provided so many equations in the ms that were already known from previous studies, which make the paper tedious.
Authors wrote this "Supplementary Materials: The following supporting information can be downloaded at: www.mdpi.com/xxx/s1" in their revised ms. I am not sure what "information" is given? This should be specifically clarified ms. For example, authors should write something like this: The supporting information contains "CCC", "DDDD" and "EEE"' otherwise, it would not allow the readers to understand what it contains.
